# No Evidence of Temperature-Driven Antimicrobial Resistance in *Salmonella* Bacteraemia in Queensland, Australia

**DOI:** 10.3390/antibiotics14121274

**Published:** 2025-12-16

**Authors:** Naveen Manchal, Megan K. Young, Maria Eugenia Castellanos, Oyelola A. Adegboye

**Affiliations:** 1Public Health and Tropical Medicine, College of Medicine and Dentistry, James Cook University, Townsville, QLD 4811, Australia; 2Metro North Public Health Unit, Metro North Hospital and Health Service, Brisbane, QLD 4031, Australia; 3School of Medicine and Dentistry, Griffith University, Goldcoast, QLD 4215, Australia; 4School of Public Health, University of Queensland, Brisbane, QLD 4006, Australia; 5Australian Institute of Tropical Health and Medicine, James Cook University, Townsville, QLD 4811, Australia; 6Menzies School of Health Research, Charles Darwin University, Darwin, NT 0811, Australia

**Keywords:** *Salmonella* bacteraemia, climate change, ambient temperature, temperature–AMR association

## Abstract

**Background:** Antimicrobial resistance (AMR) has been predicted to worsen with rising ambient temperatures and climate change, though the causal association between temperature and antimicrobial resistance in *Salmonella* species remains unconfirmed. This study investigates the association between rising ambient temperatures and resistance to antimicrobials used to treat *Salmonella* bacteraemia in Queensland, Australia. **Methods:** Time-series analysis with distributed lag non-linear models was used to test associations between deseasonalised temperature and resistance to ampicillin, ciprofloxacin, gentamicin, and third-generation cephalosporins, adjusting for precipitation, seasonality, and temporal trends. **Results:** A total of 1012 *Salmonella* bacteraemia cases were analysed in this study. Resistance to any antibiotic occurred in 25.5% of cases (95% CI: 22.8–28.3), resistance to gentamicin in 15.4% (95% CI: 13.2–17.8), and resistance to cephalosporins in 15% (95% CI: 12.9–17.4), with variation among *Salmonella* serotypes. After adjustment, no antimicrobial resistance was significantly associated with temperature: gentamicin (RR = 1.23 per 1 °C, 95% CI: 0.57–2.65, *p* = 0.59), cephalosporins (RR = 1.19, 95% CI: 0.52–2.72, *p* = 0.68), ciprofloxacin (RR = 1.88, 95% CI: 0.29–12.03, *p* = 0.50), and ampicillin (RR = 1.93, 95% CI: 0.28–13.17, *p* = 0.50). A marginal temperature–precipitation interaction for cephalosporins, identified using GAM (*p* = 0.048), did not remain significant after multiple testing correction, nor was it robust across model specifications (GLM *p* = 0.058) or cross-validation. **Conclusions:** The findings demonstrate that climate–AMR relationships are not universal, highlighting the importance of geographic, epidemiologic, and organism contexts in these associations.

## 1. Introduction

Globally, antimicrobial resistance (AMR) and climate change are the two top public health emergencies and are closely intertwined. A systematic analysis projected that almost 8 million deaths will be attributed to AMR by 2050 [1]. In Australia, a study estimated that almost 27,705 quality-adjusted life years are lost due to five common hospital-associated resistant bacteria [2]. Health care costs also rise due to prolonged hospital stays and antibiotic treatment of resistant infections [3].

The problem of AMR is projected to increase with climate change [4]. Several mechanisms have been proposed, including increased pathogen survival and the direct effects of temperature on bacterial virulence. With increased virulence and, therefore, increased invasive infections, increased antibiotic use could also be expected to contribute to a vicious cycle.

Although the majority of *Salmonella* infections are leading causes of self-limited diarrhoea worldwide [5], invasive disease, like bacteraemia, has been associated with increasing ambient temperature [6,7,8,9,10,11,12,13,14,15,16,17,18]. The proposed mechanisms are the increased proliferation of these organisms, heat stress that compromises gut mucosal integrity, and increased exposure through contaminated food and water [19,20,21]. Antibiotics are needed to treat invasive *Salmonella* infections such as bacteraemia, especially in vulnerable populations [22].

Studies have reported an alarming rise in antimicrobial resistance in *Salmonella* species [23,24]. In 2020, the European Centre for Disease Prevention and Control (ECDC) reported that 25.4% of *Salmonella* isolates from humans and 53.6% from broiler carcasses were multidrug-resistant (MDR) [25]. An Australian study, reporting data from two states, indicated that 17% of nontyphoidal *Salmonella enterica* (NTS) isolates were resistant to at least one antimicrobial agent [26]. Similarly, fluoroquinolone (FNQ) resistance in *Salmonella* [27] is increasing, posing treatment challenges, as these classes of antimicrobials were the drugs of choice for severe infections. The mechanisms of resistance include efflux pumps, inactivating enzymes, altering drug target sites, and decreased permeability [28]. In Australia, resistant *Salmonella* is often associated with travel to Southeast Asia [26,29].

There is little evidence in the existing literature linking ambient temperature to *Salmonella* AMR. Therefore, this study investigates the association between ambient temperature and AMR to antibiotics used in *Salmonella* bacteraemia in Queensland (QLD).

## 2. Results

### 2.1. Study Population and Antimicrobial Resistance Prevalence

Between January 2010 and December 2019, 1012 laboratory-confirmed *Salmonella* isolates from Queensland were analysed. Resistance to any antibiotic was found in 25.5% of cases (95% CI: 22.8–28.3), gentamicin (GEN) resistance in 15.4% (95% CI: 13.2–17.8), and cephalosporin (CEF) resistance in 15% (95% CI: 12.9–17.4). Ciprofloxacin (CIP), (6.2%, 95% CI: 4.8–7.9) and Ampicillin (AMP), (3.5%, 95% CI: 2.4–4.8) resistance rates were lower, and MDR was uncommon at 1.5% (95% CI: 0.8–2.4). Resistance rates were similar between the sexes (females, 27.3% vs. males, 24%). Age stratification showed that infants <1 year of age had the highest burden of resistance at 31.2% (95% CI: 24.4–38.7) for any antibiotic (Table 1).

Nineteen major serotypes were identified in the dataset. *S.* Virchow (*n* = 253, 25.0%), *S.* Typhimurium (*n* = 125,12.4%), and *S.* Typhi (*n* = 98, 9.7%) were the most common serotypes. The combined proportion of the top three serotypes ranged from 53.1% to 65.5% annually (mean: 59.5%) with no clear increasing or decreasing trend across seasons or over the 10-year study period. Resistance patterns varied significantly by serotype for AMP (χ^2^ = 61.39, *p* < 0.001), GEN (χ^2^ = 46.76, *p* < 0.001), and CEF (χ^2^ = 45.63, *p* < 0.001). Figure 1 presents a heatmap illustrating the percentage of Salmonella isolates resistant to six commonly used antimicrobials across different serotypes. The overall resistance to key antibiotics was low across most serotypes. Nevertheless, *S.* Enteritidis showed elevated AMP resistance (24.0%, 95% CI: 9.4–45.1). For CEF and GEN, both *S.* Virchow and *S.* Aberdeen exhibited the highest resistance among all the studied serotypes.

### 2.2. Main Effects: Temperature and Antimicrobial Resistance

Resistance rates over the 10-year study period revealed no clear secular trends for any antimicrobial (Figure 2A). The monthly resistance rates fluctuated considerably, but the annual aggregated rates remained relatively stable. Overall, CEF and GEN resistance remained higher and more variable over time, while AMP and CIP resistance were consistently low. Sampling intensity varied from two to fifteen isolates per month (median eight), with no systematic changes over time that could confound climate associations (Figure 2B). Figure 2C shows a modest increase in CEF and GEN resistance toward 2019.

Distributed lag non-linear models (DLNMs) examining deseasonalised temperature (controlling for seasonality, precipitation, and secular trends) found no significant associations with antimicrobial resistance for any of the antimicrobials tested. For GEN (*n* = 156 events), the overall temperature effect was non-significant (Wald test *p* = 0.59 for mean temperature, *p* = 0.93 for maximum temperature). Similarly, CEF resistance (*n* = 152 events) showed no significant association with temperature (*p* = 0.75 for mean temperature, *p* = 0.99 for maximum temperature) (Appendix A).

Generalised additive models for CIP (*n* = 63) and AMP (*n* = 35) resistance also revealed no significant temperature effects (all *p* > 0.12). Raw precipitation showed non-significant positive associations across multiple models (*p* = 0.06–0.07). Given the change in the CLSI breakpoint for CIP in the study period, a sensitivity analysis was performed. Resistance rates were similar pre-revision (2010–2012: 4.5%, 95% CI: 2.2–8.2%) and post-revision (2013–2019: 6.7%, 95% CI: 5.1–8.7%), with no statistically significant difference between periods (*p* = 0.273).

The pooled “any resistance” outcome (*n* = 179 events) similarly showed no temperature effects, confirming that no general climate–resistance relationship existed across antimicrobials in this cohort. (Figure 3).

Analyses using raw (non-deseasonalised) temperature to preserve seasonal climate patterns showed no effect. CEF resistance demonstrated a temperature–precipitation interaction in the GAM with a tensor product smooth (*p* = 0.047, EDF = 3.0), although the standard GLM interaction was non-significant (*p* = 0.058). The significant interaction in the GAM model did not survive multiple Bonferroni testing corrections and was not robust across model specifications. No other antimicrobials showed significant temperature–precipitation interactions—GEN (GLM *p* = 0.17, GAM *p* = 0.14), CIP (*p* = 0.28), AMP (*p* = 0.77)—or pooled resistance (GLM *p* = 0.59, GAM *p* = 0.10). Season-stratified analyses revealed no evidence that temperature–precipitation interactions varied by season for any antimicrobial (all seasonal interaction terms, *p* > 0.15). Winter showed the largest (though non-significant) interaction coefficients for both GEN (β = 2.91, *p* = 0.31) and CEF (β = 4.33, *p* = 0.15). (Appendix A) Exploratory categorical analysis of extreme weather periods (hot–dry, hot–wet, cool–wet) revealed no statistically significant associations with resistance for any antimicrobial tested (all *p* > 0.05), consistent with our primary continuous temperature analyses.

## 3. Discussion

This 10-year ecological time-series analysis provides new insights into the temporal dynamics of AMR in *Salmonella* bacteraemia and its association with ambient temperature in Queensland. The overall resistance to key antibiotics remained relatively low and stable over time. The resistance rates noted in this study are lower than those reported in previous Australian studies [26,29]. Williamson et al. [26] reported AMP resistance rates of 15–20% and MDR rates reaching 8–10% by 2015 in non-typhoidal *Salmonella* in Australia. Ingle et al. [29] documented higher MDR rates in *Salmonella enterica* serovar 4. The lower resistance rates in our cohort can be attributed to several factors, including the focus on bacteraemia compared to gastroenteritis, geographic variation between Queensland and national patterns, and potential differences in bacteraemia-associated serotypes with varying resistance profiles. The highest resistance rates were in infants in our cohort, consistent with their susceptibility to invasive disease.

Another significant finding is that resistance patterns varied by serotype. *S.* Enteritidis showed significantly higher ampicillin resistance compared to other serotypes. This finding is consistent with the patterns observed in the Williamson study [26]. *S.* Virchow and *S.* Aberdeen had higher GEN and CEF resistance than the other serotypes. The serotype-specific resistance patterns have important implications for empirical treatment and surveillance strategies.

Using DLNM, we examined potential associations between ambient temperature and AMR prevalence, adjusting for long-term and seasonal trends. Contrary to previous multi-pathogen studies [30,31] that reported positive associations between rising temperatures and AMR, no significant relationship was observed in *Salmonella* bacteraemia cases in our study. This contrasts with the findings of our meta-analysis, which showed that extreme temperatures were positively associated with *Salmonella* bacteraemia incidence [6]. Specifically, the pooled incidence rate ratio (IRR) for the three studies that included bacteraemia from *Campylobacter* and *Salmonella* sp. was 1.05 (95% CI, 1.03, 1.06) for extreme temperature. Importantly, the meta-analysis did not analyse AMR in *Salmonella*. Although ambient temperature rise can increase invasive disease in *Salmonella*, this does not always translate to an increase in AMR. This is because AMR develops from evolutionary pressure, primarily increased antimicrobial exposure, and not just climate-driven factors.

The difference likely reflects both methodological and biological factors. Previous studies reporting a positive association have not included the rarer outcome of bacteraemia in their cohort [30,31]. There are also organism-related differences, with these studies focusing on urinary infections from Enterobacteriaceae or healthcare-related infections (where resistance is highest) from *Pseudomonas* and *Acinetobacter. Salmonella* epidemiology differs significantly from that of these pathogens, with distinct transmission-related risk factors. Moreover, because *Salmonella* persists in ecological and zoonotic reservoirs, including water, soil, animals, and food chains, these reservoirs may buffer the effects of short-term climatic fluctuations, weakening direct environmental resistance coupling compared with pathogens predominantly confined to hospital settings [32,33,34].

Methodologically, studies by Bock et al. (2022) and MacFadden et al. [30,31] did not adjust for seasonal trends in their regression models. The use of simple correlation and regression models, without accounting for secular trends, introduces potential confounding. For instance, Macfadden et al. [31] averaged 30 years of historical climate data to assess cross-sectional geographic variation in AMR rather than evaluating temporal associations. Similarly, Li et al. [35] aggregated AMR data across multiple Chinese provinces, limiting the ability to distinguish temporal effects from regional differences in prescribing patterns or environmental conditions. These design features may have inflated apparent associations between temperature and resistance.

Furthermore, the modest seasonal variation in QLD may lack the temperature extremes necessary to drive resistance selection pressure. This could explain the absence of a detectable temperature–AMR relationship in our study, consistent with Li et al.’s study in Southern China [35].

Several other studies linking ambient temperature and AMR are non-human environmental studies [36,37,38,39] that report the emergence of AMR genes in bacterial genomes. It is unclear whether this translates to resistant human infections. Notably, one study reported that higher temperatures were associated with fewer AMR genes [40]. Finally, the healthcare and regulatory environment in Australia is robust, with robust food safety regulations and antimicrobial stewardship programmes. Temperature effects documented in low-resource settings with less stringent food safety and antibiotic prescribing may not manifest in well-regulated systems.

The differential resistance patterns in *Salmonella* serotypes seen in this study are consistent with the previous literature reporting higher rates of ampicillin resistance in *S.* Enteritidis [41,42]. This has implications for both clinician awareness of antimicrobial choice and the need for serotype-specific public health surveillance strategies.

This study’s major strength lies in its methodological rigour, as evidenced by a 10-year dataset, sophisticated DLNM modelling, multiple antimicrobial classes, and explicit control for key confounders. The null findings are presented transparently, thereby contributing to an evidence synthesis that is less susceptible to publication bias.

There are a few limitations to acknowledge. Laboratory surveillance data captures only diagnosed bacteraemia, and the true burden of cases might be higher. The sample sizes for ampicillin and ciprofloxacin were small, limiting subgroup analyses. Similarly, small sample sizes for different serotypes did not allow for the analysis of climate–AMR associations by serotype. The 2012–2013 CLSI CIP breakpoint revision occurred during our study period, which affects the temporal comparability for this antimicrobial. However, this represents a standard implementation of updated clinical definitions aimed at improving the detection of clinically relevant resistance. Despite the breakpoint revision, CIP resistance rates remained similar before and after the change (*p* = 0.273), suggesting the impact on temporal comparability was minimal in our cohort. Finally, we lacked individual-level data on socioeconomic status, antibiotic consumption, and travel history, which could confound or modify the associations between climate and AMR. However, the population-level socioeconomic status remained relatively constant in QLD during the study period, and we did not observe secular trends, as would be expected if socioeconomic status drove resistance [43].

## 4. Methods

### 4.1. Study Design

This study employs an ecological time-series design using retrospective surveillance and laboratory data collected in Queensland, Australia.

### 4.2. Case Data

The primary data source was Pathology Queensland, the government-operated diagnostic service covering public healthcare facilities across Queensland. As Salmonellosis is a notifiable condition in Queensland, case notifications are also reported to the Notifiable Conditions System (NOCS). To ensure comprehensive case capture, de-identified records from both Pathology Queensland and private laboratories were accessed via the NOCS. The Statistical Services Branch (SSB) of Queensland Health conducted deterministic dataset linkage to consolidate blood culture results and relevant demographic information.

A case of *Salmonella* bacteraemia was defined as any individual with a laboratory-confirmed positive blood culture for any *Salmonella* species reported between January 1, 2010, and December 31, 2019. *Salmonella* isolates with complete antimicrobial susceptibility data, a known collection date, and geographic location were included. Duplicate isolates from the same episode were excluded. Antimicrobial susceptibility testing in Queensland laboratories is performed according to the recommendations of the National Committee on Clinical Laboratory Standards (NCCLS), using disc diffusion and automated broth microdilution methods [44]. Resistance to six key classes of antimicrobials was analysed—Aminoglycosides: gentamicin (GEN); Fluoroquinolones: ciprofloxacin (CIP); Penicillins: ampicillin (AMP), third-generation cephalosporins (CEF); Folate pathway inhibitors: trimethoprim (TMP) and tetracyclines (TET). Resistance was defined according to the Clinical and Laboratory Standards Institute (CLSI) breakpoints, current at the time of testing [44]. MDR was defined as non-susceptibility to ≥3 antimicrobial classes [45]. We identified the following relevant changes in CLSI breakpoints: (1) Ceftriaxone resistance breakpoints were revised in 2010, at the start of our study period; all isolates were tested using the revised breakpoint (≥4 µg/mL). (2) CIP susceptible breakpoints were revised to ≤0.06 µg/mL in 2012–2013 to better identify decreased susceptibility. Isolates were classified according to guidelines current at the time of testing [46].

### 4.3. Climate Data

Monthly climate data were obtained from the Australian Bureau of Meteorology (BOM) from weather stations located closest to the postcode of the residences of cases. Monthly average of daily mean temperatures (°C), maximum monthly average of daily maximum temperatures (°C), and monthly cumulative rainfall (mm) were tabulated.

### 4.4. Statistical Analysis

Individual isolate-level data were aggregated to monthly counts by collection date. For each month, we calculated the prevalence of resistance for each antimicrobial.

Climate variables were matched to resistance data by year and month, creating a merged time-series dataset with monthly observations. To distinguish the effects of temperature from seasonal cycles, we used deseasonalised variables. Extreme weather conditions were defined using the 75th percentile threshold of standardised (z-score) climate variables within our study period: high temperature: standardised temperature > 75th percentile; high precipitation: standardised precipitation > 75th percentile; hot and wet: both high temperature and high precipitation; hot and dry: high temperature with precipitation ≤ 75th percentile; cool and wet: temperature ≤ 75th percentile with high precipitation; normal: all other months (reference category) [47,48]. Months were classified into meteorological seasons: Summer (December–February), hot and wet; Autumn (March–May), cooling and transitional; Winter (June–August), cool and dry; and Spring (September–November), warming and transitional.

A tiered analytical approach based on outcome frequency was used to ensure appropriate statistical power for the main effects of temperature [49]. For antimicrobials with ≥100 resistance events (gentamicin, *n* = 156, 3rd-generation cephalosporins, *n* = 152), quasi-Poisson distributed lag non-linear models (DLNMs) were fitted to examine non-linear exposure–response relationships and delayed (lagged) effects up to 3 months. Cross-basis functions were constructed using natural cubic splines with 3 degrees of freedom (df) for temperature, 2 df for lags of 0–3 months, and 4 df for controlling for seasons and long-term trends. Overall temperature effects were assessed using Wald tests across all cross-basis coefficients. Relative risks (RR) with 95% confidence intervals (CI) were estimated at temperature percentiles (10th, 25th, 75th, 90th) relative to median temperature, representing cumulative effects over the lag period.

For antimicrobials with 30–99 events (CIP, *n* = 63, AMP, *n* = 35), quasi-Poisson generalised additive models (GAMs) without lagged effects were used. Data-driven degrees of freedom selection (3–5 df) and cyclic cubic splines for month (bs = “cc”) were used.

We used simulation-based methods (1000 Monte Carlo iterations) to estimate power to detect temperature–AMR associations under observed sample size and study conditions [50]. For our primary outcomes (GEN and CEF), we had >80% power to detect RR ≥ 1.5, which represents a clinically meaningful 50% increase in resistance per 1 °C temperature rise. For secondary outcomes (CIP and AMP), power was reduced but still adequate for detecting moderate-to-large effects of public health significance. Appendix A.

Antimicrobials with <30 events (TMP, *n* = 21, TET, *n* = 2, MDR, *n* = 15) were excluded from modelling due to insufficient statistical power, consistent with minimum sample size recommendations for time-series analysis [51].

For the secondary analysis of detecting temperature–precipitation interactions, non-deseasonalised temperature was used to preserve seasonal climate patterns that may mediate resistance mechanisms. Generalised Linear Regression Models (GLMs) were used, including mean temperature, precipitation, and seasonal interactive terms. Separate GLMs were fitted for each season. For outcomes with >100 events, GAMs with tensor-product smooths were fitted to capture non-linear interactions. Categorical indicators were used for extreme weather conditions with normal weather as the reference category. All quasi-Poisson models were checked for overdispersion, temporal autocorrelation (ACF plots of residuals), and model fit (deviance residuals, QQ plots). Quasi-Poisson family was chosen over Poisson to accommodate overdispersion common in count data.

Sensitivity analyses for the deseasonalisation method and lag structure were conducted to test the robustness of the findings. To assess whether changes to CLSI breakpoints affected temporal comparability of resistance estimates, we evaluated ciprofloxacin susceptibility before and after the 2012–2013 revision that lowered the susceptible breakpoint to ≤0.06 µg/mL. Resistance proportions were compared between the pre-revision (2010–2012) and post-revision (2013–2019) periods using two-sample tests for differences in proportions with 95% confidence intervals. Ceftriaxone did not require a sensitivity assessment because its breakpoint revision occurred in 2010, the start of our study period, and all isolates were tested under the updated criteria.

Statistical significance was set at α = 0.05 (two-tailed). All analyses were conducted in R version 4.4.3 (R Foundation for Statistical Computing, Vienna, Austria) [52], using the following packages: dlnm v2.4.7 (distributed lag models) [53], mgcv v1.9.1 (generalised additive models) [54], tidyverse v2.0.0 (data manipulation and visualisation) [55], lubridate v1.9.4 (date handling) [56], and splines v4.4.3 (basis functions) [52].

## 5. Conclusions

In conclusion, after rigorous analysis specifically designed to detect climate–AMR associations, we found no evidence that temperature influences antimicrobial resistance in *Salmonella* bacteraemia in QLD, Australia. The findings demonstrate that climate–AMR relationships are not universal, highlighting the importance of geographic and epidemiological context. Evidence-based AMR control should prioritise interventions with established efficacy for antimicrobial stewardship, infection prevention, and surveillance. 

Future research should focus on identifying contexts in which climate–AMR relationships genuinely exist, rather than those in which associations reflect confounding or chance, using the methodological standards established in this study.

### Summary

This time-series study in Queensland analysed 1012 *Salmonella* bacteraemia cases to examine links between ambient temperature and antimicrobial resistance. No significant associations were found between temperature increases and resistance to GEN, CEF, CIP, or AMP after adjustment for confounders.

## Figures and Tables

**Figure 1 antibiotics-14-01274-f001:**
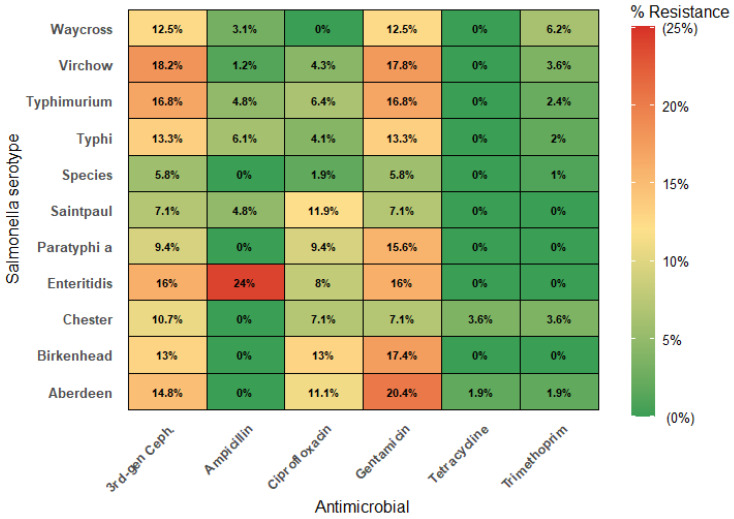
Serotype-specific antimicrobial resistance rate patterns in *Salmonella* isolates in Queensland, 2010–2019. Green indicates high antimicrobial sensitivity (low resistance), yellow indicates moderate resistance, and red indicates low sensitivity or high resistance.

**Figure 2 antibiotics-14-01274-f002:**
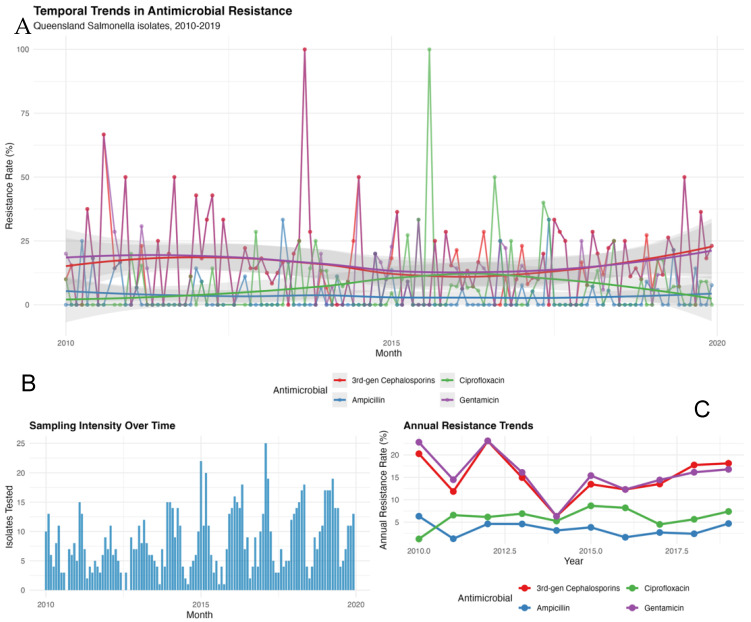
Monthly resistance rates (%) for major antimicrobials (**A**), with fitted trend lines (shaded 95% confidence intervals). (**B**) Monthly sampling intensity (number of isolates tested), and (**C**) annual mean resistance rates by antimicrobial class.

**Figure 3 antibiotics-14-01274-f003:**
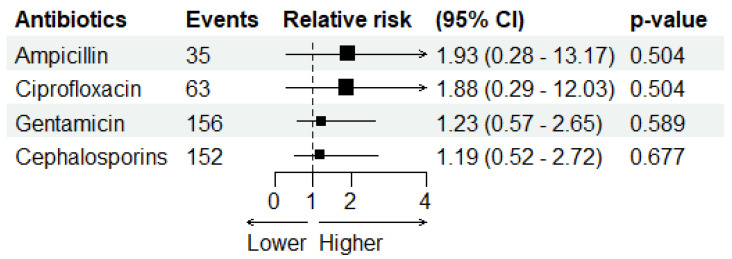
Forest plot showing the relative risk (RR) of antimicrobial resistance per 1 °C increase in deseasonalised temperature. Only antibiotics with more than 30 resistant isolates are presented. Error bars represent 95% confidence intervals. No significant associations were observed across antibiotic classes.

**Table 1 antibiotics-14-01274-t001:** Proportion of antibiotic resistance patterns of *Salmonella* isolates, overall and by sex and age group. Queensland, January 2010–December 2019.

Characteristics	Prevalence (95% CI)						
	Total Isolates (N)	AMP %	CIP %	GEN %	CEF %	Any Antibiotic %	MDR %
**Overall**	1012	3.5 (2.4–4.8)	6.2 (4.8–7.9)	15.4 (13.2–17.8)	15 (12.9–17.4)	25.5 (22.8–28.3)	1.5 (0.8–2.4)
**Sex**							
**Female**	461	3.5 (2–5.6)	7.2 (5–9.9)	16.1 (12.8–19.7)	15.6 (12.4–19.3)	27.3 (23.3–31.6)	2 (0.9–3.7)
**Male**	551	3.4 (2.1–5.3)	5.4 (3.7–7.7)	14.9 (12–18.1)	14.5 (11.7–17.7)	24 (20.4–27.7)	1.1 (0.4–2.4)
**Age group** **, years**							
**<1**	173	2.9 (0.9–6.6)	7.5 (4.1–12.5)	20.2 (14.5–27)	20.2 (14.5–27)	31.2 (24.4–38.7)	2.3 (0.6–5.8)
**1–4**	146	1.4 (0.2–4.9)	4.1 (1.5–8.7)	15.8 (10.3–22.7)	15.8 (10.3–22.7)	23.3 (16.7–31)	0.7 (0–3.8)
**5–17**	115	5.2 (1.9–11)	7 (3.1–13.2)	14.8 (8.9–22.6)	13 (7.5–20.6)	26.1 (18.3–35.1)	0.9 (0–4.7)
**18–64**	367	3.3 (1.7–5.6)	6 (3.8–8.9)	12.5 (9.3–16.4)	12 (8.8–15.8)	22.6 (18.4–27.2)	1.4 (0.4–3.2)
**≥65**	211	4.7 (2.3–8.5)	6.6 (3.7–10.9)	16.6 (11.8–22.3)	16.6 (11.8–22.3)	27 (21.1–33.5)	1.9 (0.5–4.8)

AMP—Ampicillin. CIP—Ciprofloxacin. GEN—Gentamicin. CEF—Cephalosporins. MDR—Multidrug resistant.

## Data Availability

The data that support the findings of this study are available on reasonable request from the first author. The data are not publicly available due to ethical restrictions relating to human research and the conditions of ethics approval granted by James Cook University.

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
