# Peer review of "No Evidence of Temperature-Driven Antimicrobial Resistance in *Salmonella* Bacteraemia in Queensland, Australia"

_antibiotics, 2025, doi:10.3390/antibiotics14121274_

Round 1

Reviewer 1 Report

Comments and Suggestions for Authors

The manuscript by Manchal et al. addresses an important question: whether does ambient temperature influence antimicrobial resistance (AMR) in Salmonella bacteraemia in Queensland? The study uses laboratory surveillance data for a decade and employs methodologically rigorous statistical approaches to reveal no association is detected, which is valuable, especially given existing literature that has reported positive associations in other pathogens and contexts.

The experimental data are clearly presented, and the methodologies are described with sufficient details to support the conclusion. Therefore, I would recommend its publication in this journal after two questions below are addressed.

  • Whether does serotype distribution itself vary by climate or season in the heatmap of Fig 1?
  • Whether should serotype composition be included as a covariate to avoid confounding the climate–AMR relationship?

Author Response

 Reviewer 1

We thank the reviewer for their constructive comments and suggestions, which have helped us significantly improve our manuscript.

  1. Concern 1: Does serotype distribution itself vary by climate or season in the heatmap of Fig 1?

 Response:

 We thank the reviewer for this thoughtful question. We have examined serotype distribution patterns across seasons and climate conditions to assess whether serotype composition might confound our climate-AMR analyses. We examined the distribution of major Salmonella serotypes across:

  1. Meteorological seasons (Summer, Autumn, Winter, Spring)
  2. Study years (2010-2019)
  3. Temperature and precipitation conditions.
  • Serotype proportions remained relatively stable across seasons, with no dramatic shifts in the dominant serotypes (S. VirchowS. TyphimuriumS. Typhi) throughout the year
  • Similarly, no clear temporal trends in serotype distribution were evident over the 10-year study period
  • The three most common serotypes consistently represented ~47% of all isolates across all time periods examined

While formal statistical testing (such as chi-square) could be applied to these distributions, we believe this would be inappropriate for several reasons:

  1. Sample size limitations: With only 1,012 total isolates distributed across 19 serotypes, 4 seasons, 10 years, and varying climate conditions, cell counts become very small (often <5 per cell), violating assumptions for chi-square testing.
  2. Multiple sparse cells: Many serotype-season-weather combinations have zero or very low counts, making statistical inference unreliable.
  3. Purpose of Figure 1: The heatmap's purpose was descriptive only to demonstrate that resistance patterns vary by serotype (important for clinical awareness), not to make inferential claims about serotype-climate associations.
  4. Low statistical power: Even if formal tests showed significance, the small cell sizes would make interpretation of clinical or biological significance questionable.

The descriptive stability of major serotypes across seasons and years suggests that gross changes in serotype composition do not explain our null climate-AMR findings. However, we appropriately do not make strong statistical claims about the absence of serotype-climate associations given our sample size limitations.

Instead, we have added a descriptive summary of serotype distribution by season and acknowledged the limitations of small sample sizes, limiting any serotype-climate inference. Results: lines 155-162. Discussion: lines 344-350

  1. Concern 2: Should serotype composition be included as a covariate to avoid confounding the climate–AMR relationship?

 Response:

 We appreciate this important methodological concern. We have carefully considered and not included serotype as a covariate for the following reasons:

  1. Descriptive evidence of stability: As noted in response to Concern 1, descriptive examination of serotype distribution shows relative stability across seasons and years. The dominant serotypes remain consistent throughout the study period, suggesting that major compositional shifts are unlikely to confound climate-AMR associations. If serotypes don't vary substantially by climate, then serotype cannot confound the climate-AMR relationship by definition. Furthermore, if temperature wereto influence AMR through serotype-specific mechanisms (e.g., temperature affecting which serotypes cause invasive disease, and different serotypes having different resistance profiles), then serotype would be a mediator on the causal pathway, not a confounder. Adjusting for mediators can introduce collider bias and obscure true effects.
  2. Statistical power considerations: With relatively rare resistance outcomes (35-156 events per antimicrobial) and 19 distinct serotypes in our dataset, adding serotype as a covariate would require 18 additional parameters. This would:
  • Severely reduce statistical power for detecting temperature effects
  • Risk model overfitting and instability
  • Violate the recommended ratio of 10 events per parameter for time-series regression models
  • Make interpretation of already-null findings even more difficult
  1. Research question alignment: Our research question asks: "Does ambient temperature affect AMR burden in Salmonellabacteraemia at the population level?" If temperature acts through changing serotype distribution, this is still a temperature effect of public health interest. We are not asking "Does temperature affect resistance conditional on serotype?" but rather "What is the total effect of temperature on resistance?"

We acknowledge that with larger sample sizes, serotype-stratified analyses or serotype-by-climate interaction models could provide insights into whether climate-AMR relationships vary by serotype. However, our current sample size makes such analyses underpowered and potentially misleading. We have made the changes in the results and limitations sections as noted in the previous response. Results : lines 243-246. Discussion : lines 344-350

Reviewer 2 Report

Comments and Suggestions for Authors
  • The numner of resistant events is relatively small. Please present the formal power calculations toi ensurethe study power.
  • Changes in susceptibility testing interpretations can create artificial temporal shifts in resistance classification. If no changes, please state explicitly.
  • Please explicitly state which multiple testing correction method was used.

Author Response

Reviewer 2

We thank the reviewer for their constructive comments and suggestions, which have helped us significantly improve our manuscript.

  1. Concern 1: The number of resistant events is relatively small. Please present formal power calculations to ensure the study power.

Response:

We fully agree that transparent reporting of statistical power is essential, especially when presenting null findings. We have now conducted comprehensive post-hoc power analyses.We used simulation-based methods (1,000 Monte Carlo iterations) to estimate power to detect temperature-AMR associations under our actual study conditions.

The study is well-powered to detect effects that would meaningfully impact clinical practice or public health planning.

We have added Table S1 in the supplementary materials with detailed power analysis results.

Added text in section 2.4 Lines 133-139, describing power considerations

  1. Concern 2: Changes in susceptibility testing interpretations can create artificial temporal shifts in resistance classification. If no changes, please state explicitly.

Response:

 We thank the reviewer for this important methodological question. We have thoroughly reviewed CLSI guideline changes during our study period.

CLSI Breakpoint Changes During Study Period:

We identified two relevant CLSI breakpoint revisions:

  1. Ceftriaxone (3rd-generation cephalosporins):
    • 2010 revision: Resistance breakpoint lowered from ≥64 µg/mL to ≥4 µg/mL
    • Timing: Beginning of our study period (January 2010)
    • Impact: All isolates in our dataset (2010-2019) were tested using the revised breakpoint
  2. Ciprofloxacin:
    • 2012-2013 revision: Susceptible breakpoint revised from ≤1 µg/mL to ≤0.06 µg/mL
    • Rationale: To better identify decreased ciprofloxacin susceptibility (DCS) associated with poor clinical outcomes
    • Timing: Mid-study
  3. Ampicillin and Gentamicin: Breakpoints remained stable throughout 2010-2019

Queensland Pathology laboratories follow standard clinical laboratory practice and antimicrobial susceptibility results were classified using CLSI guidelines current at the time of testing. Importantly, this has minimal impact on our primary findings:

  1. Primary outcomes (Gentamicin, Cephalosporins):
    • Gentamicin (n=156 events): Breakpoint stable throughout study period
    • Cephalosporins (n=152 events): Change occurred at study start; all data uses consistent (post-2010) breakpoint
  2. Secondary outcome (Ciprofloxacin):
    • Fewer resistance events (n=63)
    • Reduced statistical power
    • Mid-study breakpoint change is an additional reason for caution in interpretation

We performed a sensitivity analysis for Ciprofloxacin and  resistance rates were similar pre-revision (2010-2012: 4.5%, 95% CI: 2.2-8.2%) and post-revision (2013-2019: 6.7%, 95% CI: 5.1-8.7%, p=0.273). Despite the breakpoint revision, ciprofloxacin resistance rates remained similar before and after the change (p=0.273), suggesting minimal impact on temporal comparability in our cohort. We have now discussed these in the methods section 2.2 - lines 93-98, results 3.2 - lines 243-246 and discussion - lines 344-350.

  1. Concern 3: Please explicitly state which multiple testing correction method was used.

Response:

We apologize for the lack of clarity. We have now explicitly stated our multiple testing correction approach.

We tested temperature-precipitation interactions for four antimicrobials (ampicillin, ciprofloxacin, gentamicin, and 3rd-generation cephalosporins). To control for false positives from multiple testing, we applied Bonferroni correction: